# Construction of conducting bimetallic organic metal chalcogenides via selective metal metathesis and oxidation transformation

Yigang Jin[1,2], Yuhui Fang [3], Ze Li[1,2], Xiang Hao[1,2], Feng He [1], Bo Guan [1,2], Dongwei Wang[4], Sha Wu[1,2], Yang Li[1,2], Caiming Liu [1,2], Xiaojuan Dai[1,2], Ye Zou[1,2], Yimeng Sun[1,2] & Wei Xu [1,2] ✉

Conducting organic metal chalcogenides (OMCs) have attracted considerable interest for their superior electrical properties and fascinating functions. However, the electronic structural and functional regulation of OMCs are typically limited to the combination of monometallic nodes and graphene-like ligands. Here, we report a family of bimetallic OMCs ($[CuAg_x(C_6S_6)]_n$, $x = 4$ or 2) synthesized via selective metal metathesis and oxidation transformation. Both OMCs have alternatively stacked one-dimensional (1D) copper-dithiolene chains and 2D Ag-S networks, which can synergistically serve as charge transport pathways, rendering these bimetallic materials highly conductive. The incorporation of heterometallic nodes turned nonmagnetic $[Ag_5(C_6S_6)]_n$ into paramagnetic metallic $[CuAg_4(C_6S_6)]_n$, which exhibited a coherence-incoherence crossover in magnetic susceptibility measurements and an unusually large Sommerfeld coefficient, reminiscent of the characteristics of Kondo lattice. This work opens up an avenue for tailoring the electronic structures of OMCs and provides a platform for studying the dichotomy between electronic localization and itinerancy.

Organic metal chalcogenides (OMCs) represent a burgeoning class of organic–inorganic hybrid materials constructed with continuous M-X (X = S, Se, Te) networks that are further covalently connected by organic ligands and extend into periodic one-dimensional (1D), two-dimensional (2D), or three-dimensional (3D) frameworks[1–4]. Conducting OMCs have currently received significant attention for their tailorable structures[5,6] and superior electrical properties[6–8] as well as fascinating features of quantum phenomena[9,10]. Growing efforts have been dedicated to exploring conducting OMCs by varying the organic and metal building blocks. To date, by combining single-metal nodes

(e.g., Fe, Ni, Cu, or Ag, etc.) and highly symmetrical aromatic ligands, a large number of conductive OMCs with varying structural topologies have been produced[11,12]. However, these monometallic OMCs generally impose some unavoidable disadvantages in manipulating their electronic structures and regulating the functions of interest, thus limiting the maximal potential of this emerging class of organic–inorganic hybrid materials.

From the structural perspective, the integration of bimetallic nodes inside OMCs could be an effective method to tailor the targeting structural topologies and introduce additional functionalities.

[1]Beijing National Laboratory for Molecular Sciences, CAS Key Laboratory of Organic Solids, Institute of Chemistry, Chinese Academy of Sciences, Beijing 100190, China. [2]University of Chinese Academy of Sciences, Beijing 100049, China. [3]Beijing National Laboratory of Molecular Science, Beijing Key Laboratory of Magnetoelectric Materials and Devices, College of Chemistry and Molecular Engineering, Peking University, Beijing 100871, China. [4]CAS Key Laboratory of Standardization and Measurement for Nanotechnology, National Center for Nanoscience and Technology, Beijing 100190, China. ✉e-mail: wxu@iccas.ac.cn

However, the kinetic inertness and fast reaction rate of the metal–chalcogen bond formation reaction[13] make the direct synthesis impractical, since the reaction of ligands with different metal ions would inevitably suffer from the separate growth of OMCs in solution. Encouragingly, post-synthetic modification provides an alternative scheme for accessing materials when the direct synthesis route fails[14,15]. For example, in analogous systems like transition metal dichalcogenides (TMD), doping of TM (Fe, Co, and Ni) atoms has been experimentally demonstrated to be a promising way to achieve functional programming[16,17]. However, unlike the ionic TMD, OMC is a covalent compound and its skeleton is maintained by robust covalent M-X bonds[12,18], which makes the energy barrier of embedding a second metal atom into pristine OMC lattices very steep. Consequently, the synthesis of bimetallic OMCs has been challenging and has not been reported hitherto.

Here, the original bimetallic OMC, [CuAg₄(C₆S₆)]ₙ (CuAg₄BHT, BHT = benzenehexathiol), was synthesized via selective metal metathesis of Ag₅BHT, wherein only the square-planar Ag ions were precisely substituted by Cu ions. In particular, under the delicate oxidation regulation, CuAg₄BHT can be converted to another bimetallic species, CuAg₂BHT. The crystal structures of both OMCs were solved with atomic resolution, revealing 3D lamellar structures composed of alternatively stacked 1D copper-dithiolene chains and 2D Ag-S networks. Significantly, ultraviolet-photoelectron spectroscopy (UPS) and band structure calculations evidence the intrinsic metallic character of both materials. In addition, combining magnetic susceptibility and specific heat measurements, a coherence-incoherence crossover was observed in CuAg₄BHT, suggesting that this OMC would be an unprecedented candidate for a heavy Fermi liquid with Kondo lattice behavior. Our findings open up an avenue for the design of bimetallic OMCs with unusual structural topologies and tailor-made functionalities, and provide an attractive platform to search for exotic states of matter.

## Results

### Synthesis and structural characterization

As shown in Fig. 1, both the direct synthesis and post-synthetic metathesis were attempted to synthesize heterometallic OMCs. Under homogeneous solution condition, when BHT reacted with a mixture of the two metal ions, no crystalline product could be isolated (Supplementary Table 1 and Supplementary Fig. 1), due to the divergent OMC formation kinetics of two metal ions with BHT. To achieve controlled incorporation of two metal nodes, we envisioned that the post-synthetic metathesis approach with highly crystalline Ag₅BHT (ref. 19) and Cu₄BHT (ref. 20) as the parent frameworks might work, since these two well-defined OMCs possess similar building blocks and coordination geometries (Fig. 1). We screened various synthetic parameters, including temperature, reaction time, stoichiometry, and metal salt types to find optimal conditions. However, due to the appearance of silver metal, the transmetalation reactions of Cu₄BHT are always unsuccessful (Supplementary Note 1). Under optimized conditions, the reaction of Ag₅BHT with Cu(NO₃)₂·3H₂O could afford a highly crystalline product, CuAg₄BHT (Supplementary Table 2 and Supplementary Fig. 2).

Previous studies have shown that the structural transformation achieved by chemical oxidation was a synthetic shortcut to prepare OMCs[21]. To testify the universality of this chemical transformation strategy for OMCs, valence-variable cerium ammonium nitrate ((NH₄)₂Ce(NO₃)₆, CAN) was employed as the oxidizing reagent here (Methods, Supplementary Fig. 6). When two equivalents of CAN were added to CuAg₄BHT, we found that CuAg₄BHT can be completely transformed into another bimetallic species, CuAg₂BHT (Fig. 1), which is evidenced by powder X-ray diffraction (PXRD) and other structural characterizations (see the following section for details).

X-ray photoelectron spectroscopy (XPS) reveals that C, S, Cu, and Ag present within both samples (Supplementary Fig. 7), indicating the absence of solvent molecules or other counterions. The

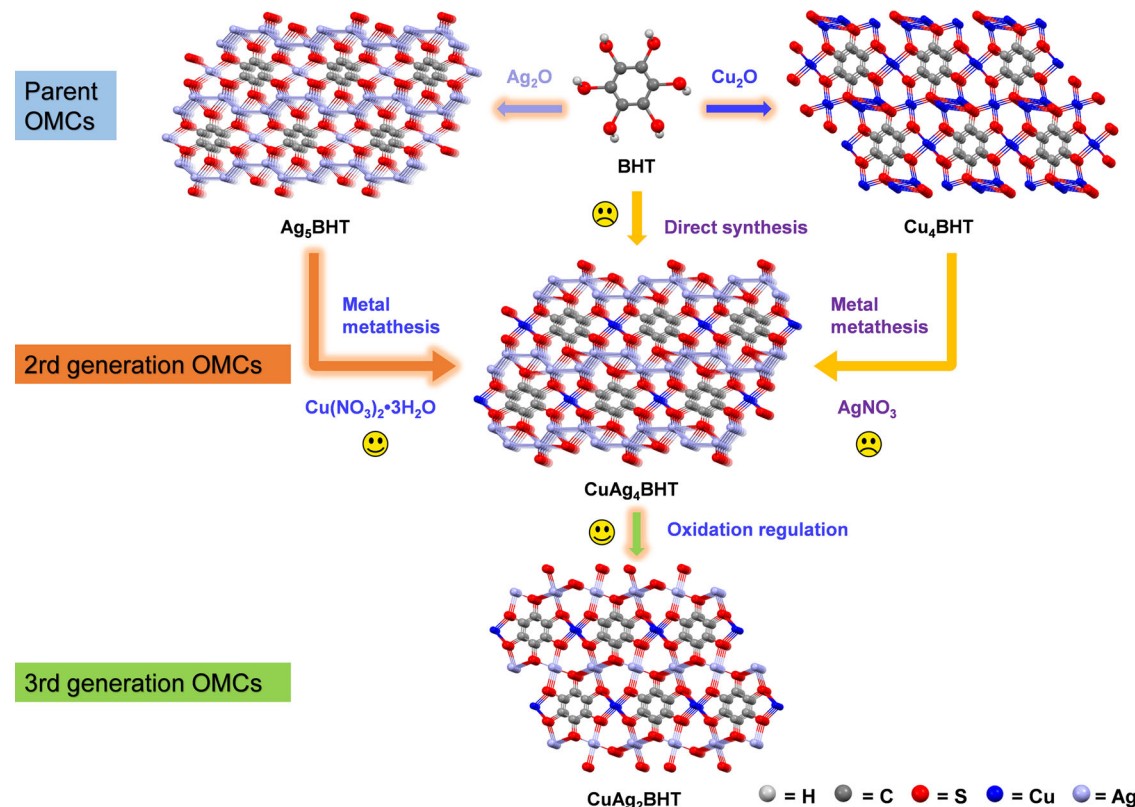

**Fig. 1 | Synthetic scheme of bimetallic OMCs.** Illustration of design strategies and synthetic routes of CuAgₓBHTs (x = 4, 2).

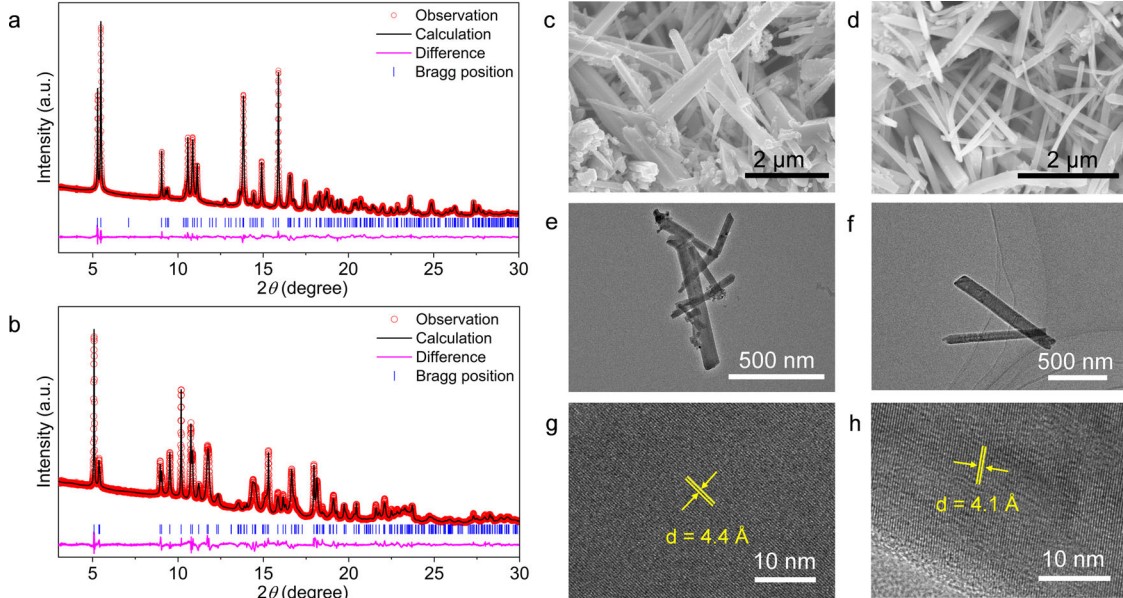

**Fig. 2 | The structural and morphology characterization of CuAg$_x$BHT.**
**a**, **b** Rietveld refinement of PXRD patterns for CuAg$_4$BHT (**a**) and CuAg$_2$BHT (**b**) from a synchrotron source ($\lambda = 0.69003$ Å). **c–f** The SEM images of CuAg$_4$BHT (**c**) and CuAg$_2$BHT (**d**). The TEM images of CuAg$_4$BHT (**e**) and CuAg$_2$BHT (**f**). **g**, **h** HRTEM images of CuAg$_4$BHT (**g**) and CuAg$_2$BHT (**h**). Source data are provided as a Source Data file.

thermogravimetric analysis shows that CuAg$_x$BHTs are stable up to 300 °C and solvent molecules are absent in the final products (Supplementary Fig. 8). The energy dispersive spectroscopy disclosed the uniform distribution of S, Cu, and Ag across the samples (Supplementary Fig. 9), suggesting that the samples prepared herein were not physical mixtures of two monometallic OMCs. The inductively coupled plasma optical emission spectrometry (ICP-OES) and Electron Probe Micro-analyzer (EPMA) results indicated that the atomic ratios of Cu:Ag:S in CuAg$_4$BHT and CuAg$_2$BHT are close to 1:4:6 and 1:2:6, respectively (Supplementary Fig. 4). Consequently, their chemical formulas can be estimated as [CuAg$_4$(C$_6$S$_6$)]$_n$ and [CuAg$_2$(C$_6$S$_6$)]$_n$, respectively.

Both CuAg$_x$BHT samples display rod-like morphologies with lengths varying from 500 nm to over 2 μm and widths around 50 nm, as shown in scanning electron microscopy (SEM) and transmission electron microscopy (TEM) images (Fig. 2c–f). In the high-resolution TEM images of CuAg$_4$BHT and CuAg$_2$BHT nanorods (Fig. 2g, h), two series of lattice fringes with interlayer distances of 4.4 and 4.1 Å were clearly discerned, suggesting the high crystallinity of CuAg$_x$BHT nanorods, which is verified by the PXRD observations (Fig. 2a, b).

The high-resolution PXRD data of CuAg$_4$BHT obtained with synchrotron radiation displayed sharp diffraction peaks from $2\theta = 5°$ to 30° with the $d$ spacings similar to what observed in Ag$_5$BHT (Supplementary Table 3), showing that these two materials are structurally similar. As the PXRD data show a lot of possibilities for the space group of this material, rotation electron diffraction (RED) was performed on CuAg$_4$BHT microcrystals to determine the cell parameter and space group accurately. First, a set of initial unit cells was obtained from the 3D reciprocal lattice reconstructed from the RED data (Supplementary Fig. 10). The results show that CuAg$_4$BHT crystallizes in the monoclinic $P2_1/c$ space group with cell parameters of $a = 4.28$ Å, $b = 8.75$ Å, $c = 14.48$ Å, and $\beta = 94.3°$. Based on this initial unit cell parameters and synchrotron PXRD data, the structure of CuAg$_4$BHT was solved with the charge flipping algorithm (a detailed description of the structure-solving process is presented in Supplementary Note 2). The final Rietveld refinement converges with $R_p = 3.57\%$, $R_{wp} = 4.64\%$. The crystal structure of CuAg$_2$BHT was directly solved based on synchrotron PXRD data with the final Rietveld refinement converging with $R_p = 2.21\%$, $R_{wp} = 2.62\%$. CuAg$_2$BHT crystallizes in a triclinic $P-1$ space group with cell parameters of $a = 3.58$ Å, $b = 8.43$ Å, $c = 8.80$ Å, $\alpha = 62.1°$, $\beta = 86.1°$, and $\gamma = 80.8°$. The small differences between the experimental PXRD patterns and the simulated ones (Fig. 2a, b) confirm the accuracy of the crystal structures of CuAg$_x$BHT. The detailed crystallography data are provided in Table 1 and Supplementary Table 4. In Fig. 2a, b, no diffraction peaks belonging to crystalline impurities could be observed, and the calculated results are in good agreement with the experimentally observed PXRD data, verifying the phase purity of CuAg$_x$BHT samples. In addition, the EPMA, ICP-OES and elemental analysis results of CuAg$_x$BHT powder samples are highly consistent with the formulas obtained based on crystal structure analysis and further confirmed the purity unambiguously (Supplementary Table 5).

Due to the shorter Cu-S bond lengths (2.247(8), 2.214(8) Å) in CuAg$_4$BHT, the substitution of the square-planar Ag atoms leads to an obvious shrink in cell volume for CuAg$_4$BHT (540.9 Å$^3$) compared to that of Ag$_5$BHT (551.7 Å$^3$) (Table 1). This also results in a reduction in the crystal symmetry from the $I2/m$ space group to $P2_1/c$, while the structural topology is almost intact. Thus, similar to that observed in Ag$_5$BHT, CuAg$_4$BHT exhibits alternatively stacked 2D Ag-S networks and 1D copper-dithiolene chains (Fig. 3a). Moreover, a graphene-like

**Table 1 | Crystallographic details of Ag$_5$BHT (ref. [19]) and CuAg$_x$BHT obtained from the Rietveld refinement using synchrotron PXRD and RED data**

|  | **Ag$_5$BHT** | **CuAg$_4$BHT** | **CuAg$_2$BHT** |
|---|---|---|---|
| Formula | Ag$_5$C$_6$S$_6$ | CuAg$_4$C$_6$S$_6$ | CuAg$_2$C$_6$S$_6$ |
| Formula weight | 803.77 | 759.44 | 543.71 |
| Crystal system | Monoclinic | Monoclinic | Triclinic |
| Space group | $I2/m$ | $P2_1/c$ | $P-1$ |
| $a, b, c$ (Å) | 15.1168, 9.1042, 4.2776 | 4.2773, 8.7547, 14.4847 | 3.5790, 8.4259, 8.8001 |
| $\alpha, \beta, \gamma$ (°) | 90, 110.43, 90 | 90, 94.31, 90 | 62.09, 86.14, 80.80 |
| $V$ (Å$^3$) | 551.67 | 540.86 | 231.49 |
| $R_{wp}$, $R_p$ | 4.98%, 3.45% | 4.64%, 3.57% | 2.62%, 2.21% |
| GOF | 3.46 | 1.42 | 1.31 |

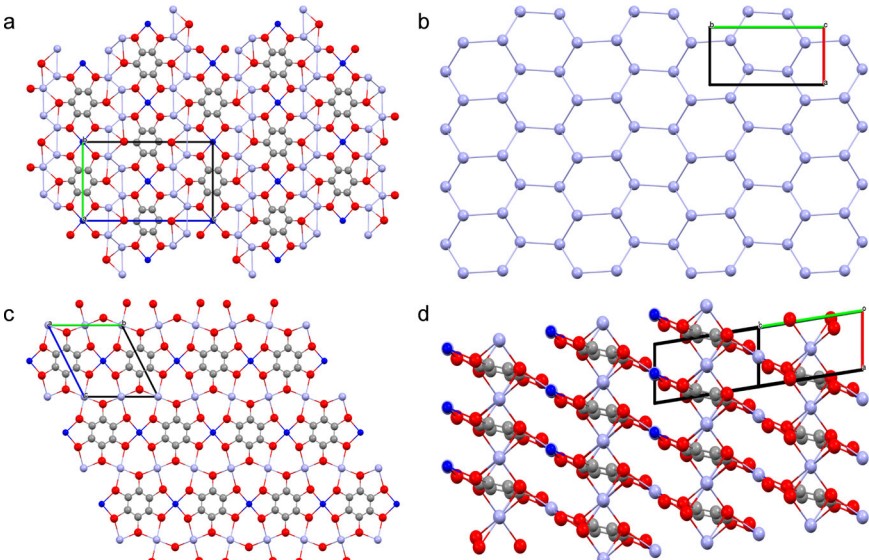

**Fig. 3 | Representation of the crystal structures of CuAg$_x$BHTs. a** Crystal structure of CuAg$_4$BHT viewed along the *a*-axis direction. **b** Honeycomb-like lattice formed by adjacent Ag atoms in CuAg$_4$BHT (viewed along the *c*-axis). **c** Top view of the Kagome-like lattice in CuAg$_2$BHT (viewed along the *a*-axis). **d** Side view of the layered structure formed by the metals ions with square-planar geometry and the C$_6$S$_6$ moieties in CuAg$_2$BHT (gray: carbon atoms, red: sulfur atoms, blue: copper atoms, violet: silver atoms).

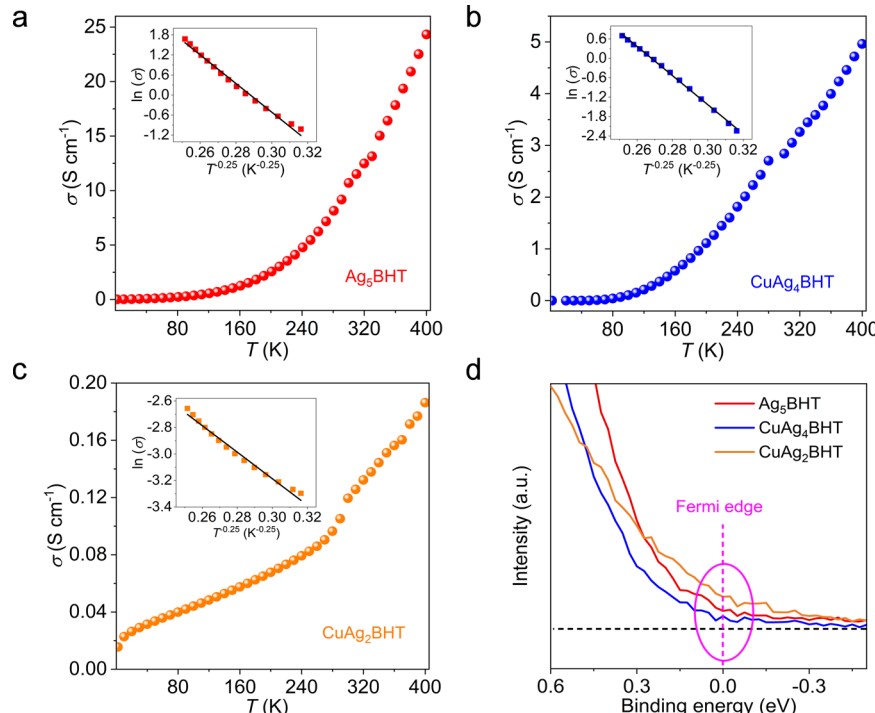

**Fig. 4 | Electrical transport characterizations. a–c** Electrical conductivities of three materials as a function of temperature from 2 to 400 K. Inset, plots of ln($\sigma$) versus $T^{-1/4}$ over the temperature region of 100–250 K. **d** UPS data for the valence band region of three samples acquired at 300 K, and the Fermi edge was shown by the pink dotted line. Source data are provided as a Source Data file.

layer structure formed by the connection of Ag atoms is preserved in the transmetalation product (Fig. 3b), which is further confirmed by the characteristic symmetric vibration signal of Ag-Ag bond in Raman spectrum (Supplementary Fig. 12). In contrast, CuAg$_2$BHT exhibits a Kagome-like lattice when viewed along the *a*-axis (Fig. 3c), in which the six-coordinated Ag atoms lied between the mean planes defined by the two square-planar metal ions and ligand moieties (Fig. 3d). It is worth noting that the Ag atoms with high coordination numbers are critical for extending bimetallic OMCs into 3D networks (Supplementary Fig. 13 and Fig. 1).

## Electrical transport characterizations

The electrical conductivities of Ag$_5$BHT and CuAg$_x$BHT samples were measured on pressed pellets via the four-probe method. At 300 K, all these OMCs exhibited high conductivities, ranging from 10 S cm$^{-1}$ for Ag$_5$BHT and 0.13 S cm$^{-1}$ for CuAg$_2$BHT to 3 S cm$^{-1}$ for CuAg$_4$BHT

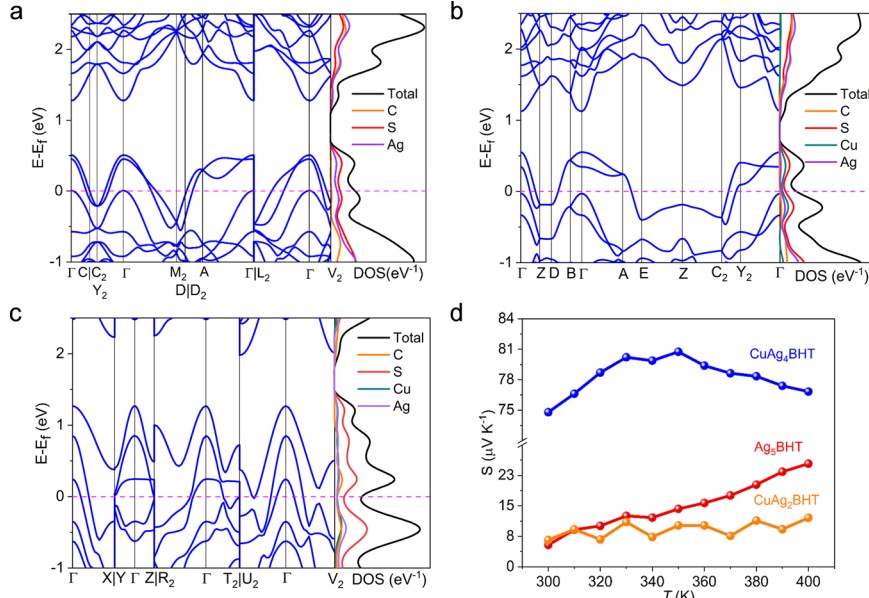

**Fig. 5 | Calculated electronic structures and Seebeck coefficient measurements.** **a–c** Calculated band structures and DOS for Ag$_5$BHT (**a**), CuAg$_4$BHT (**b**), and CuAg$_2$BHT (**c**). The high-symmetry K-points in the first Brillouin zone are provided in Supplementary Table 8. **d** Seebeck coefficients of three materials as a function of temperature from 300 to 400 K. Source data are provided as a Source Data file.

(Fig. 4a–c and Supplementary Table 6). These conductivities are among the highest values ever reported for conductive coordination polymers (Supplementary Table 7). The conductivity is positively correlated with temperature, suggesting the thermally activated transport. The plot of ln($\sigma$) versus $T^{-0.25}$ in the temperature range of 100–250 K is well fitted to the 3D Mott variable-range hopping model (Fig. 4a–c, inset)[22]. This behavior is very similar to what is observed in other granular conductors where inter-grain hopping dominates the charge transport[19,23]. Given that all three samples are composed of highly crystalline nanorods, the observed behaviors reflect inter-particle rather than intraparticle (i.e., intrinsic) transport behavior.

UPS indicated that the Fermi levels of all three samples cut the valence bands (Fig. 4d), which is an inherent character for metallic states, similar to conducting Ni$_3$BHT$_2$ and Cu$_3$(HIB)$_2$ (HIB = hexaiminobenzene)[6,24]. Density functional theory (DFT) calculations were performed to probe their electronic structures and Density of States (DOS) near the Fermi level. As shown in Fig. 5, they all exhibit wide energy band dispersions and salient features with bands crossing the Fermi level, further showing the metallic character of these materials.

The Seebeck coefficients of all three materials displayed a positive sign (Fig. 5d), indicating that hole-type conduction is dominant[25,26]. The room-temperature Seebeck coefficient of CuAg$_4$BHT is ~74 $\mu$V K$^{-1}$, which is more than ten times larger than those of CuAg$_2$BHT (-6 $\mu$V K$^{-1}$) and Ag$_5$BHT (-5 $\mu$V K$^{-1}$). According to Mott's formula, the Seebeck coefficient is connected with the variation of the DOS at the Fermi level[27,28], which means that the steeper varying DOS around the Fermi level will result in a larger Seebeck coefficient. Thus, for CuAg$_2$BHT, the observed small Seebeck coefficient could be attributed to its smoothly changing DOS near the Fermi level, while the DOS is close to its minimum (Fig. 5c). Compared to Ag$_5$BHT, the DOS of CuAg$_4$BHT displays a steeper curve around the Fermi level (Supplementary Fig. 15), consistent with the markedly larger Seebeck coefficient of CuAg$_4$BHT. Besides, we carried out theoretical calculations of the Seebeck coefficients using the Boltzmann transport theory with the constant scattering approximation as implemented in the BoltzTrap code[29–31]. With the constant scattering approximation, the Seebeck coefficients are estimated based on the band structures. The simulation results show

that the exact Seebeck values are slightly different from the experimental ones, but the trend is consistent with what observed that the Seebeck value of CuAg$_4$BHT is significantly higher than that of the other two materials (Supplementary Fig. 16). Furthermore, the valence bands of Ag$_5$BHT and CuAg$_2$BHT are mostly parabolic bands, while CuAg$_4$BHT displays nearly flat band along the E−Z−C$_2$ vectors, which is also a tag for the higher Seebeck coefficient. Notably, a bipolaron effect[32] could be observed in CuAg$_4$BHT, with the Seebeck coefficient reaching a maximum value (-81 $\mu$V K$^{-1}$) at 350 K and decreasing in the temperature range from 350 to 400 K, suggesting the contribution of the minority carriers increased with raising temperature.

As shown in Supplementary Fig. 17a–d, for monometallic Ag$_5$BHT, the DOS near the Fermi level mainly derived from the Ag-$d$ orbitals and S-$p$ orbitals, suggesting that the 2D Ag-S network is the preferred charge transport route. Due to the different coordination geometries and chemical environments of Ag1 and Ag2 in Ag$_5$BHT (Ag1 refers to the six-coordinated Ag atoms with distorted octahedron geometry and Ag2 features a square-planar coordination mode), the $d$-orbitals splitting of Ag1 and Ag2 is expected to be different, which makes the orbital compositions of Ag1 and Ag2 different in PDOS. The Ag1$_{d_{xz}+d_{x^2-y^2}}$ and Ag2$_{d_{xy}+d_{yz}}$ orbitals contribute the most to PDOS near the Fermi level (Supplementary Fig. 17c, d), whereas for CuAg$_x$BHTs, due to copper incorporating into the framework matrix, the PDOS near the Fermi level is dominated by contributions from the $d$ orbitals of Cu and Ag and $p$ orbitals of S. It should be noted that for CuAg$_4$BHT, the PDOS derived from Ag $d$-orbitals is different from that of Ag1 in Ag$_5$BHT, while the PDOS contribution from Cu$_{d_{xy}+d_{yz}}$ orbitals is similar to that of Ag2 in Ag$_5$BHT. The $p$-orbital composition of S atoms also changes after the metal atom substitution (Supplementary Fig. 17b, f), which is ascribed to the charge redistribution in this M$_5$BHT system (M refers to the metal ions). After the chemical oxidation transformation, significant variations in the PDOS composition of the $d$ orbitals of metal atoms and the $p$ orbitals of C and S atoms can be observed near the Fermi level (Supplementary Fig. 17i–l). All these results elucidate that the introduction of bimetallic nodes can synergistically tailor the electronic structures of the OMCs, thereby modulating the corresponding charge transport behavior.

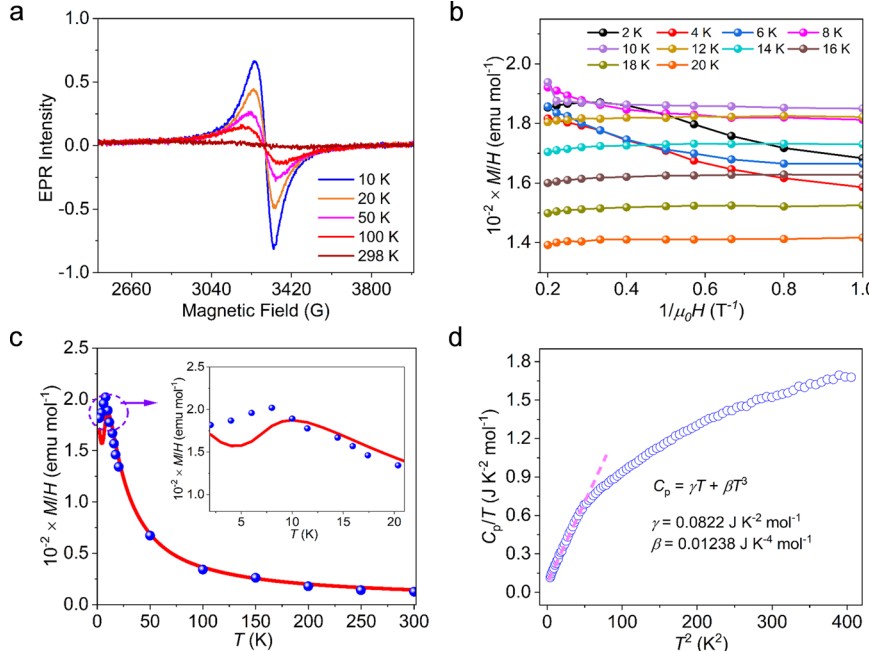

**Fig. 6 | Temperature dependence of various physical properties of CuAg4BHT.** **a** Temperature variable EPR spectrum. **b** $M(H)|_T$ isotherm measurements performed at various temperatures. As argued by Johnston, $M(H)|_T = \chi(T)H + M_i(H)|_T$, where $\chi(T)$ and $M_i(H)|_T$ are the intrinsic susceptibility of CuAg4BHT and the impurity contribution to $M(H)|_T$, respectively. **c** Measured magnetic susceptibility $M/H$ of CuAg$_4$BHT (red solid line). The blue symbols show the intrinsic susceptibilities $\chi(T)$ of CuAg$_4$BHT derived from magnetic isotherms. The upper inset shows $M/H$ versus $T$ plot below 20 K. **d** $C_p/T$-$T^2$ plot of CuAg$_4$BHT in the temperature range of 2–20 K. The dashed line represents a fit using the equation, $C_p = \gamma T + \beta T^3$. Source data are provided as a Source Data file.

## EPR and magnetic studies

Electron paramagnetic resonance (EPR) measurements were performed to investigate electron configurations of these materials. For Ag$_5$BHT, no EPR signal was detected even when cooled down to 10 K, suggesting its diamagnetic nature. After metal metathesis, a nearly isotropic signal contributed by the Cu$^{2+}$ with $g = 2.053$ was observed (Fig. 6a), which is consistent with the Cu 2$p$ XPS analysis in CuAg$_4$BHT (Supplementary Fig. 18). As for CuAg$_2$BHT, only a very weak EPR signal was observed (Supplementary Fig. 19). The signal intensity is temperature dependent and almost diminishes at room temperature, which indicates that it does not originate from an organic radical. As this EPR signal was broad and asymmetrical, it can be attributed to a small amount of leftover Cu$^{2+}$ during oxidation, while the dominant Cu ions became EPR silent, supporting that the oxidation state of Cu in CuAg$_2$BHT was +1 (Supplementary Fig. 18).

From the first generation of Ag$_5$BHT (coordination assembly), to the second generation of CuAg$_4$BHT (metal metathesis), and then to the third generation of CuAg$_2$BHT (oxidation regulation) (Fig. 1), the non-innocent ligand shows multiple accessible redox states that are responsible for electrically neutral networks. During the metathesis process, we found that only the square-planar Ag atoms of Ag$_5$BHT could be replaced by Cu(II) ions, even by raising the reaction temperature or prolonging the duration. Generally, longer bond length refers to lower bond energy, so it can be expected that the bond energy of Ag-S bond (231.3–202.1 KJ mol$^{-1}$) is lower than that of Cu-S bond (274.5 KJ mol$^{-1}$)[33]. Thus, the precise exchange result can be attributed to the labile tetra-coordinated [AgS$_4$] cores (Supplementary Fig. 13), which has also been verified by other studies[34,35], whereas Ag atoms with non-coplanar coordination geometry were stabilized by the formation of Ag-Ag bond, preventing their substitution by Cu atoms. Furthermore, the total energy of Ag$_5$BHT (−122.4 eV) calculated with DFT method is higher than that of CuAg$_4$BHT (−125.6 eV), indicating this selective metal metathesis reaction is a thermodynamically favored process.

The precise substitution of the Ag$^+$ ions by Cu$^{2+}$ ions turns the nonmagnetic Ag$_5$BHT into a paramagnetic conducting material. The coexistence of local spins and conducting electrons reminds us of the research on molecular Kondo system[36]. Remarkably, searching for Kondo systems in 3$d$-electron compounds has attracted immense interest in condensed matter physics[37]. Considering that Cu$^{2+}$ ions with localized magnetic moments are periodically arranged in the network having itinerant $d$-$p$-$\pi$ electrons, Kondo lattice scenario arising from the interaction between localized spins and conduction electrons in CuAg$_4$BHT could be expected.

Thus, the magnetic susceptibility measurements were carried out first. As shown in Fig. 6c, $M/H$ increases with decreasing temperatures and displays a broad peak around 12 K. A significant downward trend occurs as the temperature decreases, and the $M(T)|_H$ curve starts to increase again below 5 K. This upturn is typical of heavy Fermi liquids and reveals the presence of a small amount of magnetic or paramagnetic impurities/defects[38,39]. Actually, we noticed such paramagnetic impurities originating from the surface contamination or defects, and a calibration procedure was carried out to eliminate the interference signal. Meanwhile, in order to ensure the dominant source of magnetic properties of CuAg$_4$BHT, the possible influence of extrinsic oxides was excluded by synchrotron PXRD and temperature variable EPR measurements (Supplementary Figs. 19 and 20). Therefore, the intrinsic magnetic susceptibility of CuAg$_4$BHT was precisely determined by analyzing the magnetic isotherms $M(H)|_T$ at different temperatures (Fig. 6b) instead of $M(T)|_H$ curve[40]. The derived intrinsic susceptibilities were represented by the blue symbols in Fig. 6c, and followed the Curie–Weiss law in the high $T$ region. The Curie–Weiss fitting from 100 to 300 K afforded a Curie constant of $C = 0.376$ emu K mol$^{-1}$ and Curie–Weiss temperature of $\theta = -5.6$ K. The calculated effective moment ($\mu_{eff}$) for each Cu ion in the formula CuAg$_4$C$_6$S$_6$ equals 1.74 $\mu_B$, which was reasonably close to that expected for Cu(II) (S = 1/2, $\mu_{eff} = 1.73$ $\mu_B$). The negative Curie–Weiss temperature indicated the

presence of antiferromagnetic interactions between the magnetic moments. Besides, the measured susceptibility shows no signature of a magnetic ordering transition down to 2 K, but a clear deviation from Curie–Weiss behavior appears below $T < 12$ K. As the temperature decreases, the intrinsic susceptibility of CuAg$_4$BHT reaches a broad hump around 10 K and then decreases when $T < 8$ K, while its intrinsic susceptibility remains quite large, equal to $1.8 \times 10^{-2}$ emu mol$^{-1}$ toward the $T = 0$ limit. Such magnitude and $T$ dependence of susceptibility[39,41] provides a crude estimation of the coherence temperature $T^*$ (refs. [42], [43]). Below the $T^*$ of -8–10 K, the susceptibility evolving becomes visible, indicating the emergence of the interaction of local Cu$^{2+}$ spins and conduction $d$-$p$-$\pi$ electrons. Therefore, the observable decrease in intrinsic susceptibility when $T < 8$ K is attributed to the Kondo scenario, in which localized magnetic moments are screened by itinerant electrons. Thus, with decreasing temperature, a crossover from a paramagnetic state with localized moments to renormalized Fermi liquid state with coherent Kondo lattice was suggested in CuAg$_4$BHT.

In addition, since CuAg$_4$BHT is isostructural with Ag$_5$BHT, the comparison of their specific heat data can reveal the influence of Cu ion introduction. In the temperature range from 2 to 8 K, the specific heat data are well represented by the relation of $C_p = \gamma T + \beta T^3$ (Fig. 6d). A linear fit of $C_p/T$ to $T^2$ plot yields a Sommerfeld coefficient ($\gamma$) of 82.2 mJ K$^{-2}$ mol$^{-1}$ for CuAg$_4$BHT, which is more than ten times higher than that of Ag$_5$BHT (Supplementary Fig. 21). Such unusually large $\gamma$ value also proves the existence of heavy-mass quasiparticle states[44,45], supporting the observed susceptibility anomaly. Furthermore, anomalously large Seebeck coefficient in metallic CuAg$_4$BHT provides another hint for the existence of a large effective electron mass. Finally, we realized that CuAg$_4$BHT has a triangular lattice of Cu(II) cations (Supplementary Fig. 22), the origin of the heavy quasiparticle mass could be closely linked with the geometrical frustration, as also demonstrated in $d$-electron LiV$_2$O$_4$ spinel[41]. To gain in-depth insight into the complex Kondo physics, the preparation of bulk single-crystal sample (>50 μm) and ultralow temperature measurements are further demanded, and will be performed on this OMC in the near future.

## Discussion

In summary, we have rationally designed and synthesized a family of bimetallic OMCs, CuAg$_x$BHTs, via selective metal metathesis and delicate oxidation regulation. Both crystal structures have been well characterized by PXRD and RED techniques, revealing atomically precise structures composed of alternatively stacked 1D copper-dithiolene chains and 2D Ag-S networks. As suggested by band structure calculations, the bimetal-bis(dithiolene) moieties can synergistically regulate the electronic structures of these OMCs, which accounts for the high electrical conductivity observed in 3D OMCs. In addition, the susceptibility anomaly and other measurements suggested that CuAg$_4$BHT was a candidate for a heavy Fermi liquid with Kondo lattice. We expect that our experimental results will motivate theoretical efforts on this intriguing subject in the future. More importantly, the development of bimetallic OMCs may effectively avoid the "Buckets Effect" caused by the unbalanced properties of their monometallic counterparts. Our results pave the way for constructing organic–inorganic hybrid materials with unique structural topologies, and unlock more opportunities for condensed matter physics as well as electronics, catalysis, and energy-related applications.

## Methods
### Synthesis of CuAg$_4$BHT
Under argon atmosphere, Cu(NO$_3$)$_2$·3H$_2$O (470 mg, 1.94 mmol) was dissolved in 160 mL degassed ethanol, and then Ag$_5$BHT (100 mg, 0.124 mmol) was added. The mixture was sonicated for 5 min, and

subsequently heated to 80 °C with constant stirring for 72 h to form a gray-black powder, followed by natural cooling to room temperature. The obtained power was filtered, washed with water, ethanol, acetone, and diethyl ether in sequence, and then dried at 80 °C under vacuum for 24 h. Yield: 82 mg (87%). Elem. Anal. Calcd. for CuAg$_4$C$_6$S$_6$: C, 9.49; S, 25.33; Cu, 8.37, Ag, 56.81. Found: C, 9.79; S, 25.46; Cu, 9.20, Ag, 55.55.

### Trials for the oxidation regulation of CuAg$_4$BHT
Under argon atmosphere, different amounts of Ce(NH$_4$)$_2$(NO$_3$)$_6$ (0.5–6 equiv. to CuAg$_4$BHT) were dissolved in 100 mL degassed acetonitrile, and then CuAg$_4$BHT (76 mg, 0.1 mmol) was added. The mixture was stirred at room temperature for 12 h. The obtained power was filtered, washed with water, CH$_3$CN, acetone, and diethyl ether in sequence, and then dried at 80 °C under vacuum for 24 h.

### Optimized synthetic condition of CuAg$_2$BHT
Under argon atmosphere, Ce(NH$_4$)$_2$(NO$_3$)$_6$ (220 mg, 0.4 mmol) was dissolved in 150 mL degassed acetonitrile, and then CuAg$_4$BHT (152 mg, 0.2 mmol) was added. The mixture was stirred at room temperature for 24 h. The obtained power was filtered, washed with water, CH$_3$CN, acetone, and diethyl ether in sequence, and then dried at 80 °C under vacuum for 24 h. Yield: 48 mg (88%). Elem. Anal. Calcd. for CuAg$_2$C$_6$S$_6$: C, 13.25; S, 35.38; Cu, 11.69, Ag, 39.68. Found: C, 14.53; S, 36.16; Cu, 10.23, Ag, 38.48.

### SEM and TEM characterizations
SEM images were obtained using a Toshiba SU8000-SEM with an acceleration voltage of 10 kV. TEM images were acquired using a JEOL 2100F-TEM with an acceleration voltage of 120 kV. About 1 mg of the sample was dispersed in 4 mL of absolute ethanol, and after ultrasonic dispersion for 30 min, 5 μL of the suspension was dropped on a copper grid coated with a carbon film to perform TEM characterizations. The 3D RED data were collected on a Themis 300 TEM by a BM-Ceta camera with an acceleration voltage of 300 kV. One typical set of RED data was collected by combing specimen tilt and electron-beam tilt in the range of −65° to +70° with a beam-tilt step of 1°. The nanocrystal selected for data collection was isolated by the selected area aperture, and calibrated to a concentric height to stay within the aperture over the entire tilt range. During continuous rotation, the dose rate was calibrated to <0.03 e$^-$/Å$^2$ s. Altogether, 137 frames were obtained, and the reflection intensities and unit parameters were extracted with the software Pets2.0.

### Electrical property measurement
The electrical conductivities were measured on the pelletized samples via a four-probe method in a Physical Property Measurement Systems (PPMS-9, Quantum design). About 5 mg samples were compressed into a cuboid pellet under a pressure of 10 MPa. Then four parallel gold electrodes were deposited on the pellet by vacuum evaporation to form a four-electrode device. The Seebeck coefficients were measured using the SB-100 Seebeck Measurement System (MMR Tech.) under a vacuum environment below 18 mTorr.

### Details of physical property measurements
The DC magnetic susceptibility measurement under 2–300 K was conducted in a Superconducting Quantum Interference Device (SQUID) magnetometer (Quantum Design). Zero field cooling and field cooling measurements were performed under an applied field of 1000 Oe. Diamagnetic correction was applied using Pascal's constant. The temperature variable EPR spectrum were collected using a Bruker ElexSys E580 spectrometer, operating at the X-band (ω = 9.37 GHz). The center field was set to 300 mT, and the scanning range was 100 mT. A low-temperature environment was achieved by using an Oxford Instruments ESR900 liquid helium cryostat. The low-

temperature heat capacity measurements were also performed in PPMS.

## Data availability

The crystallographic information obtained by Rietveld refinement of PXRD data has been deposited in the Cambridge Crystallographic Data Centre under accession codes CCDC 2143107 (CuAg$_4$BHT) and CCDC 2143106 (CuAg$_2$BHT) [https://www.ccdc.cam.ac.uk/structures/?]. All data generated and analyzed in this study are included in the article and its Supplementary Information, and are also available from the authors upon request. The source data generated in this study are provided in the Source Data file. Source Data are provided with this paper.

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

## Acknowledgements

The authors acknowledge the financial support from the National Key R&D Program of China (Grant No. 2017YFA0204701), the National Natural Science Foundation of China (Grant 22071256), and the Chinese Academy of Sciences (QYZDY-SSW-SLH024). The authors also appreciate the crew of the BL14B1 beamline at the Shanghai Synchrotron Radiation Facility (SSRF) for their assistance in synchrotron PXRD data collection.

## Author contributions

W.X. planned and designed the research project. Y.G.J. executed the syntheses, chemical, spectroscopic and electrical characterization. Y.H.F. performed and analyzed temperature variable EPR data. X.H. and B.G. obtained and analyzed the transmission electron microscopy data. C.M.L. performed and analyzed the magnetic susceptibility measurements. Y.L., Z.L., and S.W. collected the PXRD data. Y.M.S. contributed to the Seebeck coefficient measurements and analysis. X.J.D. and Y.Z. performed and analyzed electrical conductivity experiments and UPS characterizations. D.W.W. performed and analyzed the specific heat data. F.H. and W.X. performed the DFT calculation and crystal structure-solving works. Y.G.J. and W.X. wrote the manuscript. All authors participated in the discussion and commented on the manuscript.

## Competing interests

The authors declare no competing interests.

## Additional information

**Correspondence and requests** for materials should be addressed to Wei Xu.

