## [Peer Review File · Nature Communications]

REVIEWER COMMENTS

Reviewer #1 (Remarks to the Author):

This manuscript reports the first family of bimetallic OMCs synthesized via selective metal metathesis and oxidation transformation. The bimetal-bis(dithiolene) moieties can synergistically regulate the electronic structures of these OMCs. These findings open up a new avenue for the design of bimetallic OMCs with unusual structural topologies and tailor-made functionalities, and provide an attractive platform to search for exotic states of matter. I suggest accepting it. Some issues listed below should be solved before publication:

1. Please provide a more detailed explanation for the main contribution sources of DOS near the Fermi level.
2. Table 6, please supplement the data of other coordination polymers with high conductivity.
3. Please explain the relationship between intrinsic susceptibility and temperature in more details. From Fig. 6b, it is clear that when $T < 8$ K, the intrinsic susceptibility decreases with the decrease of temperature.
4. The color of the text in Figure 1 (c, d) is too close to the background color, please correct it.
5. The labeling direction of the lattice fringes in Figure 1h is inconsistent with the direction of the lattice fringes, please correct it.
6. Some important tables and figure are suggested to place in the text rather than in the supplementary documents, so that the article can be read more smoothly and better understood.

Reviewer #2 (Remarks to the Author):

The manuscript by Jin et al reported the first synthesis of a conducting bimetallic organic metal chalcogenides (OMCs), $[\text{CuAg}_4(\text{C}_6\text{S}_6)]_n$, via selective metal metathesis of $[\text{Ag}_5(\text{C}_6\text{S}_6)]_n$. Further delicate oxidation regulation would convert it into another new bimetallic species, $[\text{CuAg}_2(\text{C}_6\text{S}_6)]_n$. However, many critical issues required further consideration.

1. Detailed analysis of the PXRD peaks in Fig. 2a and 2b should be discussed to determine the crystal structures of $[\text{CuAg}_4(\text{C}_6\text{S}_6)]_n$ and $[\text{CuAg}_2(\text{C}_6\text{S}_6)]_n$.
2. The authors should comment on the purity and stability of the products. From the XPS spectra present in Figure S7, O 1s peak is visible in the $[\text{Ag}_5(\text{C}_6\text{S}_6)]_n$ sample, which becomes pronounced in $[\text{CuAg}_4(\text{C}_6\text{S}_6)]_n$ and outstanding in $[\text{CuAg}_2(\text{C}_6\text{S}_6)]_n$ (even much higher than the Ag 3p peak). Combined with Fig. S13 and S14, it is undoubtedly that the samples are significantly oxidized, including sulfur oxides, copper oxides and so on.
3. As the $[\text{CuAg}_2(\text{C}_6\text{S}_6)]_n$ might be significantly oxidized, the measured electronic and magnetic properties could originate from the extrinsic oxides, but not the intrinsic properties. In particular, direct evidence of the intrinsic magnetism of the $[\text{CuAg}_4(\text{C}_6\text{S}_6)]_n$ is lacking.
4. Based on the electrical transport characterizations shown in Fig. 4, $[\text{Ag}_5(\text{C}_6\text{S}_6)]_n$ sample possesses the highest conductivity, which degrades in $[\text{CuAg}_4(\text{C}_6\text{S}_6)]_n$ and $[\text{CuAg}_2(\text{C}_6\text{S}_6)]_n$. It seems that the successful synthesis of the bimetallic OMCs, however, sacrifice the conductivity.
5. The explanation of the steeper varying DOS around the Fermi level for $[\text{CuAg}_4(\text{C}_6\text{S}_6)]_n$ is also tricky. Actually, it is difficult to identify which one is steeper, $[\text{CuAg}_4(\text{C}_6\text{S}_6)]_n$ or $[\text{CuAg}_2(\text{C}_6\text{S}_6)]_n$, from the calculated electronic structures shown in Fig. 5. However, based on the UPS spectra in Fig. 4, it seems that $[\text{CuAg}_4(\text{C}_6\text{S}_6)]_n$ is more flatten than $[\text{CuAg}_2(\text{C}_6\text{S}_6)]_n$ around the Fermi level.

RESPONSE TO REVIEWERS' COMMENTS

Reviewer #1:

General Comment: This manuscript reports the first family of bimetallic OMCs synthesized via selective metal metathesis and oxidation transformation. The bimetal-bis(dithiolene) moieties can synergistically regulate the electronic structures of these OMCs. These findings open up a new avenue for the design of bimetallic OMCs with unusual structural topologies and tailor-made functionalities, and provide an attractive platform to search for exotic states of matter. I suggest accepting it. Some issues listed below should be solved before publication:

Response: We appreciate the Reviewer#1's encouraging comments and the positive recommendation. According to these valuable suggestions, we have revised the manuscript to improve the quality of our work. We hope that our additional efforts in the revised manuscript address your concerns appropriately.

Comment 1: Please provide a more detailed explanation for the main contribution sources of DOS near the Fermi level.

Response: We appreciate your constructive suggestions. In the revised SI, partial density of states (PDOS) of the three materials studied were presented (Supplementary Fig. 17). Further discussion on the difference of PDOS near the Fermi level was added, which helps us to elucidate electronic structure variation induced by the metal metathesis and oxidation conversion.

Revisions:

As shown in Supplementary Fig. 17a-d, for monometallic Ag₅BHT, the DOS near the Fermi level mainly derived from the Ag-*d* orbitals and S-*p* orbitals, suggesting that the 2D Ag-S network is the preferred charge transport route. Due to the different coordination geometries and chemical environments of Ag1 and Ag2 in Ag₅BHT (Ag1 refers to the six-coordinated Ag atoms with distorted octahedron geometry and Ag2 features a square-planar coordination mode), the *d*-orbitals splitting of Ag1 and Ag2 is expected to be different, which makes the orbital compositions of Ag1 and Ag2 different in PDOS. The Ag1_{*d*_{xz}+*d*_{x²-y²}} and Ag2_{*d*_{xy}+*d*_{yz}} orbitals contribute the most to PDOS near the Fermi level (Supplementary Fig. 17c,d). Whereas, for CuAg_xBHTs, due to copper incorporating into the framework matrix, the PDOS near the Fermi level is dominated by contributions from the *d* orbitals of Cu and Ag and *p* orbitals of S. It should be noted that for CuAg₄BHT, the PDOS derived from Ag *d*-orbitals is different from that of Ag1 in Ag₅BHT, while the PDOS contribution from Cu_{*d*_{xy}+*d*_{yz}} orbitals is similar to that of Ag2 in Ag₅BHT. The *p*-orbital composition of S atoms also changes after the metal atom substitution (Supplementary Fig. 17 b and f), which is ascribed to the charge redistribution in this M₅BHT system (M refers to the metal ions). After the chemical oxidation transformation, significant variations in the PDOS composition of the *d* orbitals of metal atoms and the *p* orbitals of C and S atoms can be observed near the Fermi level (Supplementary Fig. 17i-l). All these results clearly demonstrate the electronic structure modulation induced by the metal metathesis and chemical oxidation conversion.

Supplementary Figure 17. Partial density of states (PDOS) of Ag₅BHT (a-d), CuAg₄BHT (e-h) and CuAg₂BHT (i-l).

Comment 2: Table S6, please supplement the data of other coordination polymers with high conductivity.

Response: As suggested by the reviewer, we have supplemented conductivity values of other coordination polymers with high conductivity, which are provided as Supplementary Table 7 in the revised SI.

Comment 3: Please explain the relationship between intrinsic susceptibility and temperature in more details. From Fig. 6b, it is clear that when $T < 8$ K, the intrinsic susceptibility decreases with the decrease of temperature.

Response: Thanks for your helpful suggestion. To clarify the relationship between intrinsic susceptibility and temperature more clearly, in the revised main text, a detailed interpretation on why the intrinsic susceptibility decreases with the decrease of temperature when $T < 8$ K and some important literatures (refs. 39, 41, 42, 43) were added.

Revisions:

Besides, the measured susceptibility shows no signature of a magnetic ordering transition down to 2 K, but a clear deviation from Curie–Weiss behavior appears below $T < 12$ K. As the temperature decreases, the intrinsic susceptibility of CuAg₄BHT reaches a broad hump around 10 K and then decreases when $T < 8$ K, while its intrinsic susceptibility remains quite large, equal to 1.8×10^{-2} emu mol⁻¹ towards the $T = 0$ limit. Such magnitude and T dependence of susceptibility^{39,41} provides a crude estimation of the coherence temperature T^* (ref.^{42,43}). Below the T^* of ~8-10 K, the susceptibility evolving becomes visible, indicating the emergence of the interaction of local Cu²⁺ spins and conduction d - p - π electrons. Therefore, the observable decrease in intrinsic susceptibility when $T < 8$ K are attributed to the Kondo scenario, in which localized magnetic moments are screened by itinerant electrons.

Comment 4: The color of the text in Figure 2 (c, d) is too close to the background color, please correct it.

Response: Thanks for the suggestion. We have changed the color of the text into black in the revised Figure 2 (c, d) to make it look clearer.

Comment 5: The labeling direction of the lattice fringes in Figure 2h is inconsistent with the direction of the lattice fringes, please correct it.

Response: Correction has been made in the revised Figure 2h. Thanks for pointing out this error.

Comment 6: Some important tables and figure are suggested to place in the text rather than in the supplementary documents, so that the article can be read more smoothly and better understood.

Response: Supplementary Table 3, Supplementary Figs. 16 and 17 in our previous version have been moved to the revised main text (revised as Table 1 and Fig. 6). Thanks for your helpful suggestion to make our work more readable.

Reviewer #2:

General Comment: The manuscript by Jin et al reported the first synthesis of a conducting bimetallic organic metal chalcogenides (OMCs), $[\text{CuAg}_4(\text{C}_6\text{S}_6)]_n$, via selective metal metathesis of $[\text{Ag}_5(\text{C}_6\text{S}_6)]_n$. Further delicate oxidation regulation would convert it into another new bimetallic species, $[\text{CuAg}_2(\text{C}_6\text{S}_6)]_n$. However, many critical issues required further consideration.

Response: We appreciate the Reviewer #2 for the valuable scientific comments, which drives us to further analyze the structure/composition and the magnetic/electrical properties in the current work, thus building up a reliable structure-property relationship. According to your valuable suggestions, we have further explored the magnetic properties and supplied the additional data to ensure the dominant source of the magnetic properties. We hope our additional efforts appropriately address your concerns.

Comment 1: Detailed analysis of the PXRD peaks in Fig. 2a and 2b should be discussed to determine the crystal structures of $[\text{CuAg}_4(\text{C}_6\text{S}_6)]_n$ and $[\text{CuAg}_2(\text{C}_6\text{S}_6)]_n$.

Response: We appreciate your constructive suggestions. Here, we added some discussions addressing the similarity of the PXRD patterns of Ag_5BHT and CuAg_4BHT , as well as why RED was employed to determine the cell parameters and space group of CuAg_4BHT . In order to determine the crystal structures of CuAg_xBHTs , a series of software had been used to analyze the PXRD data, including the EXPO2014 for index, Jana2006 for profile fitting and structure solving as well as Rietveld refinement. The detailed descriptions of the

structure solving procedure are presented in the Section 2 of the Supplementary Information. In the main text, the crystal structures obtained were discussed in detail.

Revisions:

The following description of PXRD data and structure solving was added in the revised main text:

The high-resolution PXRD data of CuAg₄BHT obtained with synchrotron radiation ($\lambda = 0.69003 \text{ \AA}$) displayed sharp diffraction peaks from $2\theta = 5^\circ$ to 30° with the d spacings similar to what observed in Ag₅BHT (Supplementary Table 3), showing that these two materials are structurally similar. As the PXRD data show a lot of possibilities of the space group of this material, rotation electron diffraction (RED) was performed on CuAg₄BHT microcrystals to determine the cell parameter and space group accurately.

Supplementary Table 3. The comparison of some similar d spacings of Ag₅BHT and CuAg₄BHT obtained from PXRD patterns.

Ag ₅ BHT				CuAg ₄ BHT			
(hkl)	d spacing (Å)	(hkl)	d spacing (Å)	(hkl)	d spacing (Å)	(hkl)	d spacing (Å)
(110)	7.6673	($\bar{1}$ 21)	3.1168	(011)	7.4868	($\bar{1}$ 13)	3.1033
(200)	7.0898	(130)	2.9708	(002)	7.2219	(121)	2.9567
($\bar{1}$ 01)	4.2719	(211)	2.9041	(100)	4.2652	(113)	2.9052
(310)	4.1959	($\bar{5}$ 01)	2.8276	(021)	4.1892	(031)	2.8604
(220)	3.8337	(420)	2.7981	(110)	3.8343	(024)	2.7855
($\bar{2}$ 11)	3.7937	(510)	2.7079	($\bar{1}$ 02)	3.7997	(032)	2.7057
($\bar{3}$ 01)	3.7785	(301)	2.6382	($\bar{1}$ 11)	3.7697	($\bar{1}$ 23)	2.6446
(011)	3.6728	(330)	2.5558	(111)	3.6454	(114)	2.5443
(101)	3.5502	($\bar{2}$ 31)	2.4561	(102)	3.5573	(033)	2.4956
(400)	3.5549	($\bar{5}$ 21)	2.4027	($\bar{1}$ 12)	3.4856	(006)	2.4073

Comment 2: The authors should comment on the purity and stability of the products. From the XPS spectra present in Figure S7, O 1s peak is visible in the [Ag₅(C₆S₆)]_n sample, which becomes pronounced in [CuAg₄(C₆S₆)]_n and outstanding in [CuAg₂(C₆S₆)]_n (even much higher than the Ag 3p peak). Combined with Fig. S13 and S14, it is undoubtedly that the samples are significantly oxidized, including sulfur oxides, copper oxides and so on.

Response: Thanks for the comments on the purity and stability issues. All of these OMCs have high phase purity. Firstly, the diffraction patterns of these samples were indexed using N-TREOR09 program integrated in EXPO2014 package based on the first 20 intensive

peaks, which gave a credible unit cell with the final Rietveld refinement converging with $R_p = 3.45\%$, 3.57% , 2.21% for Ag_5BHT , $CuAg_4BHT$ and $CuAg_2BHT$, respectively. As shown in Fig. 2a,b and Supplementary Fig. 3, no diffraction peaks belonging to crystalline impurities could be observed, and the calculated results are in good agreement with the experimentally observed PXRD data, verifying the phase purity of Ag_5BHT and $CuAg_xBHT$ samples. Besides, in our previous version, EPMA characterizations provided atomic ratios of Cu: Ag: S approximately 1 : 4.25 : 5.78 for $CuAg_4BHT$ and 1 : 1.67 : 5.78 for $CuAg_2BHT$ (Supplementary Table 4), which agrees well with the expected $CuAg_xBHT$ formulas. In addition, in order to check the validity of the obtained formulas, we have performed elemental analysis by a combination of ICP-OES and C, H, N, S combustion method. The ICP-OES and elemental analysis results of $CuAg_xBHT$ powder samples are highly consistent with the formulas obtained based on crystal structure analysis and further confirmed the purity unambiguously (Supplementary Table 5).

In order to check the stability of the attained products, TGA was carried out by a TGA Q500 instrument with a heating rate of $10\text{ }^\circ\text{C min}^{-1}$. As shown in Supplementary Fig. 8, all these materials show the thermal stability up to $300\text{ }^\circ\text{C}$, indicating that no solvent molecules are contained in the final products. Besides, PXRD patterns of Ag_5BHT and $CuAg_xBHT$ samples were further studied. It was found that no significant changes in the PXRD patterns were observed after they were exposed to ambient air for 3 months. Therefore, these products were believed to have good chemical stability.

Revisions:

The following data and compositional analysis were added in the revised manuscript:

All of these OMCs have high phase purity. Firstly, the diffraction patterns of three samples were indexed using N-TREOR09 program integrated in EXPO2014 package based on the first 20 intensive peaks, which gave a credible unit cell with the final Rietveld refinement converging with $R_p = 3.45\%$, 3.57% , 2.21% for Ag_5BHT , $CuAg_4BHT$ and $CuAg_2BHT$, respectively. As shown in Figs. 2a,b and Supplementary Fig. 3, no diffraction peaks belonging to crystalline impurities could be observed, and the calculated results are in good agreement with the experimentally observed PXRD data, verifying the phase purity of Ag_5BHT and $CuAg_xBHT$ samples. Besides, EPMA characterizations provided atomic ratios of Cu: Ag: S approximately 1: 4.25: 5.78 for $CuAg_4BHT$ and 1: 1.67: 5.78 for $CuAg_2BHT$ (Supplementary Fig. 4), which agrees well with the expected $CuAg_xBHT$ formulas. In addition, in order to check the validity of the obtained formulas, we have performed elemental analysis by a combination of ICP-OES and C, H, N, S combustion method. The ICP-OES and elemental analysis results of

CuAg_xBHT powder samples are highly consistent with the formulas obtained based on crystal structure analysis and further confirmed the purity unambiguously (Supplementary Table 5).

Supplementary Table 5. Elemental analysis results of Ag₅BHT and CuAg_xBHT samples.

Formula	C (wt%)		S (wt%)		Cu (wt%)		Ag (wt%)	
	E ^a	T ^b	E ^a	T ^b	E ^a	T ^b	E ^a	T ^b
Ag ₅ C ₆ S ₆	9.12	8.97	23.32	23.93	/	/	67.43	67.10
CuAg ₄ C ₆ S ₆	9.79	9.49	25.46	25.33	9.20	8.37	55.55	56.81
CuAg ₂ C ₆ S ₆	14.53	13.25	36.16	35.38	10.23	11.69	38.48	39.68

^a Experimental results. ^b Theoretical values.

Supplementary Figure 11. The comparison of PXRD patterns of Ag₅BHT (a), CuAg₄BHT (b), CuAg₂BHT (c) samples before and after three months of air exposure. It was found that no significant changes in the PXRD patterns were observed after they were exposed to ambient air for 3 months. Therefore, these products were believed to have good chemical stability.

We fully understand the reviewers' concerns about the presence of oxide impurities in the products, which are also critical research motivations for our team in controlled synthetic chemistry. In Supplementary Fig. 7 (revised as Supplementary Fig. 18), the O 1s peaks at ~531.8 eV can be ascribed to surface-adsorbed oxygen species (adsorbed from the solvent or from the air atmosphere)¹³⁻¹⁵, which is general in the other organic metal chalcogenides¹⁶⁻²⁰. As illustrated in Fig. 1, from the 1st generation of Ag₅BHT (coordination assembly), to the 2nd generation of CuAg₄BHT (metal metathesis), and then to the 3rd generation of CuAg₂BHT (oxidation regulation), the deployment of post-synthesis methods gradually increases the defect states in the attained products, which has also been discussed in other bimetallic MOFs^{21,22}. These stepwise processes inevitably provide more coordination unsaturated sites for the adsorption of oxygen species and result in a more significant O 1s peak. In the high-resolution S 2p spectrum (Supplementary Fig.

18), the discernable S 2p peak at ~167.9 eV in CuAg_xBHTs indicates that adsorbed oxygen strongly interacts with sulfur forming the sulfate or sulfite-like species, which can often be found in the other dithiolene-type compounds^{16,31,39-41}. Nonetheless, because XPS is a surface analysis technique with a detection depth of about 3-10 nm, it is clear that there is some oxygen species presenting on the surface rather than the bulk samples, as also evidenced by the phase purity analysis discussed above.

Revisions:

These detailed discussions about XPS data were added in the revised manuscript.

Comment 3: As the [CuAg₂(C₆S₆)]_n might be significantly oxidized, the measured electronic and magnetic properties could originate from the extrinsic oxides, but not the intrinsic properties. In particular, direct evidence of the intrinsic magnetism of the [CuAg₄(C₆S₆)]_n is lacking.

Response: Thanks for the comments. First of all, as discussed in the response to Comment 2, the bulk compositions of CuAg_xBHT samples have been characterized by XPS, synchrotron PXRD, calculated results, EPMA and elemental analysis, which unambiguously confirms the dominant source of electronic and magnetic properties of these materials. But it should be noticed that the conductive and magnetic behaviors presented here did not exclude the influence of the defects that may arise from surface states, grain boundaries, the crystallite tilting and the edge states of these materials, due to the polycrystalline feature and structural complexity as a result of heterogeneity. In fact, we noticed a small amount of paramagnetic impurities originating from the surface contamination or defects, and a calibration procedure was carried out to eliminate the interference signal. Thus, the intrinsic magnetic susceptibility of CuAg₄BHT was precisely determined by analyzing the magnetic isotherms $M(H)|_T$ at different temperatures (Fig. 6b) instead of $M(T)|_H$ curve according to Johnston's equation. Based on the derived intrinsic susceptibilities, the calculated effective moment (μ_{eff}) for each Cu ion in the formula CuAg₄C₆S₆ equals 1.74 μ_B , which agrees well with the expected value for a spin-only S = 1/2 moment ($\mu_{\text{eff}} = 1.73 \mu_B$).

To further address your concerns on the influence of the presence of the extrinsic oxides on the magnetic properties of CuAg_xBHT, we reinvestigated the PXRD patterns and

EPR signals of these samples. We have added the following discussions and data in the revised manuscript and hope our additional efforts appropriately address your concerns:

Supplementary Figure 20. The comparison of synchrotron PXRD data of $[\text{CuAg}_x(\text{C}_6\text{S}_6)]_n$ samples and copper oxide.

Firstly, the diffraction data of $[\text{CuAg}_x(\text{C}_6\text{S}_6)]_n$ were collected at synchrotron radiation facility ($\lambda = 0.69003 \text{ \AA}$), which has a high sensitivity for the identification of copper oxide impurities. The comparison of synchrotron PXRD data of $[\text{CuAg}_x(\text{C}_6\text{S}_6)]_n$ and CuO was provided in the revised SI (Supplementary Fig. 20), no diffraction peaks associated with CuO can be detected in these products.

Secondly, EPR is a powerful technique for detecting these magnetic impurities. According to J.B. Goodenough's study, the Néel temperature (T_N) of CuO is 230 K (ref.⁴²), and the T_N of Cu_4O_3 is about 40 K (ref.⁴³). So, if the magnetic properties originate from the copper oxides, the EPR signal will disappear when the temperature drops below these characteristic temperatures. We conducted temperature variable EPR spectrum tests and found that no such phenomenon was observed in these two samples (Fig.6a and Supplementary Fig. 19).

Furthermore, the spin orbit coupling of Cu *d* electrons is larger than that of S or O *p* electrons, so the linewidth (ΔH_{pp}) of Cu(II) signal is much wider than that of sulfur-oxide radicals. As shown in Fig. 6a, the EPR signal with *g* value of 2.053 and ΔH_{pp} of 92 Gauss was observed in CuAg_4BHT , completely different from the narrower ΔH_{pp} (2~4 Gauss)⁴⁴⁻⁴⁶ and the smaller *g* values (~2.0051) of sulfur oxide radicals. As for CuAg_2BHT , the EPR signal intensity is temperature dependent and diminishes at room temperature, which indicates that it does not originate from an organic radical. This signal was broad (ΔH_{pp} =32 Gauss) and asymmetrical, which can be attributed to the g_{\perp} part of the anisotropic EPR spectrum of magnetically dilute Cu(II). Thus, it is assigned to a small amount of leftover Cu(II) during the oxidation. The existence of Cu(II) could be confirmed by EPR, but CuAg_2BHT was found to be

diamagnetic at room temperature, suggesting that the number of leftover Cu(II) was nearly negligible (<0.1%). The sharp contrast between CuAg₄BHT and CuAg₂BHT indicates that the measured magnetic properties does not originate from extrinsic oxides, further verifying the dominant source of magnetic properties of CuAg₄BHT.

Comment 4: Based on the electrical transport characterizations shown in Fig. 4, [Ag₅(C₆S₆)]_n sample possesses the highest conductivity, which degrades in [CuAg₄(C₆S₆)]_n and [CuAg₂(C₆S₆)]_n. It seems that the successful synthesis of the bimetallic OMCs, however, sacrifice the conductivity.

Response: Thanks for the comments. We agree with the Reviewer #2 that [CuAg_x(C₆S₆)]_n exhibited lower electrical conductivities than [Ag₅(C₆S₆)]_n, but the conductivity values of these two bimetallic OMCs are still considerable compared to conducting OMCs reported so far (Supplementary Table 7). More importantly, as discussed in the Introduction, the integration of bimetallic nodes inside OMCs could be an effective method to tailor the targeting structural topologies and introduce additional functionalities, which is a crucial research motivation for our team in this emerging field of organic-inorganic hybrid materials. What we want to show is that we cannot ignore innovative research, and exploring new topological structures and functions of OMCs is crucial in this burgeoning field. We can confidently answer the reviewer that our research is very original in both methodology and structure.

Here, CuAg₄BHT and CuAg₂BHT were successfully synthesized via selective metal metathesis and oxidation transformation, which is challenging and has not yet been realized in OMCs community. Both bimetallic OMCs feature unique topological structures composed of alternatively stacked 1D copper-dithiolene chains and 2D Ag-S networks, as revealed by PXRD and RED techniques. The measured Seebeck coefficients and band structure calculations indicate that the bimetal-bis(dithiolene) moieties can synergistically tailor the electronic structures of the OMCs, which holds great promise for modulating the corresponding charge transport behavior. Besides, the incorporation of heterometallic nodes turns nonmagnetic Ag₅BHT into paramagnetic metallic CuAg₄BHT and then to diamagnetic CuAg₂BHT, which can be ascribed to the charge redistribution induced by metal metathesis and oxidation regulation. Conclusively, these results demonstrated that the design of bimetallic OMCs is a promising manner to manipulate their electronic

structures and regulate the functions of interest, and provides an attractive platform for exploring electronic applications and the exotic quantum phenomena related to the dichotomy between electronic localization and itinerancy.

Comment 5: The explanation of the steeper varying DOS around the Fermi level for $[\text{CuAg}_4(\text{C}_6\text{S}_6)]_n$ is also tricky. Actually, it is difficult to identify which one is steeper, $[\text{CuAg}_4(\text{C}_6\text{S}_6)]_n$ or $[\text{CuAg}_2(\text{C}_6\text{S}_6)]_n$, from the calculated electronic structures shown in Fig. 5. However, based on the UPS spectra in Fig. 4, it seems that $[\text{CuAg}_4(\text{C}_6\text{S}_6)]_n$ is more flatten than $[\text{CuAg}_2(\text{C}_6\text{S}_6)]_n$ around the Fermi level.

Response: Thanks for the insightful comment from this reviewer. We agree with the referee that the difference of changing tendency of DOS near Fermi level is not so significant in Fig. 5. So, we replot the DOS of these BHT-based materials in the same figure, focusing on the changing tendency near the Fermi level (Supplementary Fig. 15). As can be clearly seen from this figure, the DOS curve of CuAg_4BHT is steeper than the other two materials. Furthermore, we carried out theoretical calculations of the Seebeck coefficients using the Boltzmann transport theory with the constant scattering approximation as implemented in the BoltzTrap code. With the constant scattering approximation, the Seebeck coefficients are estimated based on the band structures. BoltzTrap code is a well-established method for predicting the Seebeck coefficient, which has been applied to a series of thermoelectric materials and agrees well with the experimental results (*Adv. Funct. Mater.* 2008, **18**, 2880–2888; *Nat. Rev. Mater.* 2017, **2**, 17053). The simulation results show that the exact Seebeck values are slightly different from the experimental ones, but the trend is consistent with what observed that the Seebeck value of CuAg_4BHT is significantly higher than that of the other two materials (Supplementary Fig. 16). Due to the poor signal-noise ratio of UPS spectra around the Fermi level, it could give a qualitative conclusion on relative position of the Fermi level and the valance band. However, it is difficult to quantitatively analyze the slope of DOS near the Fermi level.

Revisions:

These data and discussions have been added in the revised SI and briefly mentioned in the main text.

Supplementary Figure 15. The DOS of Ag₅BHT and CuAg_xBHTs near the Fermi level.

Supplementary Figure 16. Temperature dependence of Seebeck coefficients of Ag₅BHT and CuAg_xBHT simulated via Boltzmann transport equation using Boltztrap2 code.

REVIEWERS' COMMENTS

Reviewer #1 (Remarks to the Author):

I am satisfied with the revisions and pleased to suggest accepting it.

Reviewer #2 (Remarks to the Author):

In the revised version and the response letter, much effort had been made to provide more supplementary data and detailed analyses. The authors have addressed some of my concerns. However, I am still worry about the oxidation issue of the sampes in air, and the discussion of the DOS near the Fermi level, which are relavant to the UPS measurements and the Seebeck coefficient (previous comment #5).

RESPONSE TO REVIEWERS' COMMENTS

Reviewer #1 (Remarks to the Author):

I am satisfied with the revisions and pleased to suggest accepting it.

Response: We appreciate the Reviewer#1 for the positive recommendation for publication.

Reviewer #2 (Remarks to the Author):

In the revised version and the response letter, much effort had been made to provide more supplementary data and detailed analyses. The authors have addressed some of my concerns. However, I am still worry about the oxidation issue of the samples in air, and the discussion of the DOS near the Fermi level, which are relevant to the UPS measurements and the Seebeck coefficient (previous comment #5).

Response: We gratefully appreciate for your comments. The XPS (Supplementary Fig. 7 and Supplementary Fig. 18), PXRD (Fig. 2, Supplementary Fig. 11 and Supplementary Fig. 20), elemental analysis results (Supplementary Table 5) and temperature variable EPR spectrum as well as magnetic susceptibility measurements were presented in our paper, all these results excluded the influence of the extrinsic oxides on the magnetic properties of CuAg_xBHT , and provided strong proof on the dominant source of magnetic properties of CuAg_4BHT . The relationship between the Seebeck coefficient and changing tendency of DOS near the Fermi level is well accepted in the community of thermoelectric materials, some important literatures (refs. 28-31) supporting this method have been cited in the main text with adequate discussions and a comparison illustration (Supplementary Fig. 15 and Supplementary Fig. 16). So, we are confident of our results.